# Effect of zinc deficiency on chronic kidney disease progression and effect modification by hypoalbuminemia

Atsuyuki Tokuyama[1], Eiichiro Kanda[2]*, Seiji Itano[1], Megumi Kondo[1‡], Yoshihisa Wada[1‡], Hiroyuki Kadoya[1‡], Kengo Kidokoro[1‡], Hajime Nagasu[1‡], Tamaki Sasaki[1‡], Naoki Kashihara[1‡]

1 Department of Nephrology and Hypertension, Kawasaki Medical School, Kurashiki, Okayama, Japan,
2 Medical Science, Kawasaki Medical School, Kurashiki, Okayama, Japan

☯ These authors contributed equally to this work.
‡ MK, YW, HK, KK, HN, TS and NK also contributed equally to this work.
* kms.cds.kanda@gmail.com

**Data Availability Statement:** The data cannot be fully shared publicly. The reasons are as follows: Data contain potentially sensitive information; Patients did not provide informed consent

## Abstract

Serum zinc (Zn) levels tend to be low in chronic kidney disease (CKD) patients. This cohort study was conducted to investigate the relationship between zinc deficiency and CKD progression. Patients were classified into two groups based on Zn levels < 60 μg/dl (low-Zn group, n = 160) and ≥ 60 μg/dl (high-Zn group, n = 152). The primary outcome was defined as end-stage kidney disease (ESKD) or death and was examined over a 1-year observation period. Overall, the mean Zn level was 59.6 μg/dl and the median eGFR was 20.3 ml/min/ 1.73 m$^2$. The incidence of the primary outcome was higher in the low-Zn group ($p$<0.001). Various Cox proportional hazards models adjusted for baseline characteristics showed higher risks of the primary outcome in the low-Zn group than in the high-Zn group. Competing risks analysis showed that low Zn levels were associated with ESKD but not with death. Moreover, in propensity score-matched analysis, the low-Zn group showed a higher risk of the primary outcome [adjusted hazard ratio 1.81 (95% confidence interval 1.02, 3.24)]. Furthermore, an interaction was observed between Zn and serum albumin levels (interaction $p$ = 0.026). The results of this study indicate that zinc deficiency is a risk factor for CKD progression.

## Introduction

Zinc is an essential trace element that contributes to various physiological activities in the human body. It is essential for cell survival, proliferation, and differentiation, and is required for the activation of more than 300 enzymes [1]. It is also involved in a wide variety of cellular functions as a structural constituent of numerous proteins, including enzymes involved in cellular signaling pathways and DNA replication, as well as transcription factors [1, 2]. In addition, the zinc finger and homeobox (ZHX) family of transcription factors is highly expressed in podocytes, and these factors are also involved in glomerular disease [3]. Therefore,

regarding release of personal data; the Ethics Board of Kawasaki Medical School imposed data restriction. The data are owned by Kawasaki Medical School Hospital. Interested readers may request the data at Kawasaki Medical School; URL (Japanese), https://h.kawasaki-m.ac.jp/data/contact/each. And the following Email address of the hospital may be useful for readers: hsyomu@med.kawasaki-m.ac.jp.

**Funding:** This work was supported by the Health, Labour and Welfare Sciences Research Grants, the Ministry of Health, Labour and Welfare, Japan (No. 19AC1002), the Japan Society for the Promotion of Science (KAKENHI Grant No. JP19K08740), and AMED under Grant Number JP20ek0210135. The funders had no role in study design, data collection and analysis, decision to publish, or preparation of the manuscript.

**Competing interests:** The authors have declared that no competing interests exist.

imbalances in zinc homoeostasis can lead to impairments in immune function, growth, cognitive function, and metabolism [4–7].

Serum zinc (Zn) levels tend to decrease with the progression of chronic kidney disease (CKD) and are lower in maintenance dialysis patients [8–10]. Patients on hemodialysis frequently develop zinc deficiency due to zinc removal during hemodialysis, inadequate dietary intake, and malabsorption. We previously demonstrated that zinc supplementation ameliorates the response to erythropoietin therapy in dialysis patients [11]. In diabetic rats, zinc supplementation suppresses the pathological changes associated with tubulointerstitial and glomerular damage [12]. Moreover, a clinical study demonstrated that zinc supplementation reduces urinary albumin excretion in patients with type 2 diabetes mellitus (DM) with microalbuminuria [13]. These lines of evidence suggest that zinc supplementation may protect against the exacerbation of renal dysfunction.

CKD is a significant risk factor for end-stage kidney disease (ESKD) and cardiovascular disease [14, 15]. Prevention of CKD progression is an important strategy to improve poor prognosis and avoid deterioration in quality of life. However, no studies have used hard renal endpoints to investigate whether zinc deficiency promotes CKD progression. Therefore, we conducted this retrospective cohort study using a CKD patient database of electronic medical records to investigate whether zinc deficiency is a risk factor for CKD progression using the hard renal endpoint, ESKD.

## Materials and methods

### Study design and participants

This is a retrospective cohort study using a CKD database based on electronic medical records from Kawasaki Medical School Hospital. The study protocol was approved by the Ethics Board of Kawasaki Medical School and was exempt from the need to obtain informed consent from participants (No. 3656–1). It was performed in accordance with the relevant guidelines and the Declaration of Helsinki of 1975, as revised in 1983. The data were anonymized before analysis.

We extracted 4,066 patients who visited the Department of Nephrology and Hypertension from January 1, 2014 to December 31, 2017 (Fig 1). Among them, we identified 577 patients whose Zn levels were measured during this period and whose estimated glomerular filtration rate (eGFR) was less than 60 ml/min/1.73 m$^2$. We excluded patients who had already undergone renal replacement therapy (n = 250), patients who had only a single visit (n = 2), and patients with missing data (n = 13). On the basis of these criteria, 312 patients were included in this study. Based on the "Practice Guideline for Zinc Deficiency" of the Japanese Society of Clinical Nutrition, zinc deficiency was defined as a Zn level < 60 μg/dl [2]. The patients were classified into two groups: those with Zn levels < 60 μg/dl (low-Zn group) and those with Zn levels ≥ 60 μg/dl (high-Zn group).

The extracted data were as follows: age; sex; body mass index (BMI); comorbidities; medications; eGFR; serum zinc, albumin, creatinine, and C-reactive protein (CRP) levels; hemoglobin levels; dipstick proteinuria; and date of renal replacement therapy initiation or death. Nephrotic syndrome was defined as follows: a history of primary nephrotic syndrome or meeting the diagnostic criteria of nephrotic syndrome (serum albumin level ≤ 3.0 g/dl and urinary protein/creatinine ratio ≥ 3.5 g/g Cr). Cardiovascular disease was defined as a history of ischemic heart disease, heart failure, severe valvular disease, peripheral arterial disease, cardiovascular surgery, or stroke. Chronic liver disease comprised cirrhosis and chronic hepatitis. The types of bowel disease were inflammatory bowel disease, malabsorption syndrome, and short

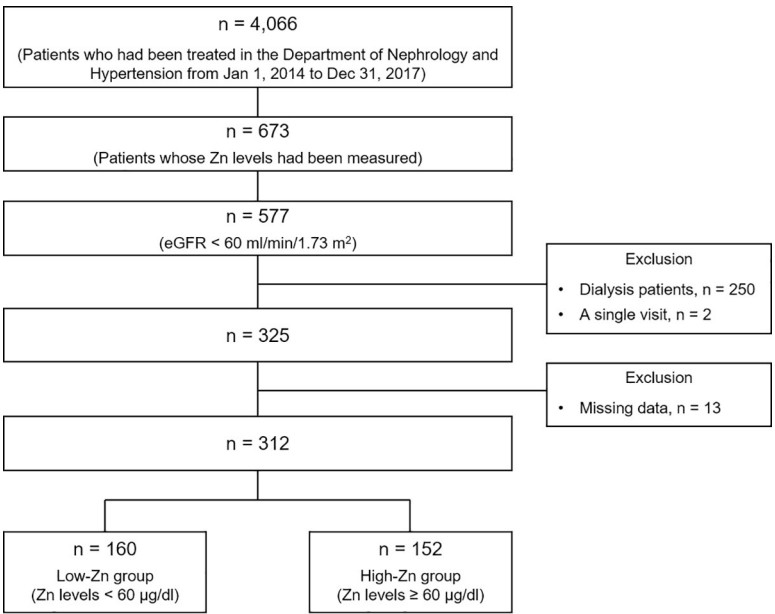

**Fig 1. Flowchart of the study population.** Of 4,066 patients, 312 were included in this study. The study patients were classified into two groups: those with Zn levels < 60 μg/dl (low-Zn group) and those with Zn levels ≥ 60 μg/dl (high-Zn group). Abbreviation: Zn, serum zinc.

bowel syndrome. Proteinuria was defined as urinary protein excretion of 1+ or more by dipstick test.

The observation period was 1 year. The primary outcome was defined as ESKD or death. If neither outcome occurred within 1 year, the observation was censored.

## Statistical analyses

Baseline characteristics are shown as the mean ± SD for those with a normal distribution; otherwise, the median and interquartile range are presented. Highly skewed variables were transformed with the natural logarithm function prior to their use in models (eGFR and CRP). Baseline patient characteristics and outcomes were compared using the chi-square test, t-test, and Mann-Whitney U test, as appropriate.

Survival rates were evaluated using Kaplan-Meier survival curves, and statistical significance was assessed using the log-rank test. The survival curves of the risk groups were then evaluated using the Kaplan-Meier survival curves. After proportional hazards assumption was confirmed using a double logarithmic plot, multivariate Cox proportional hazards models were used to compare the risks of outcomes between the groups. The results are presented here as hazard ratios (HRs) with 95% confidence interval (CI). In the analysis of the competing risks of ESKD and death, we used Fine and Gray competing risk regression models. The factors included in the Cox proportional hazards models and competing risks models were basic patient characteristics and the factors showing statistically significant differences between the low- and high-Zn groups, such as age, male sex, BMI, DM, cardiovascular disease, ln(eGFR), ln(CRP), serum albumin levels, hemoglobin levels, dipstick proteinuria, angiotensin II receptor blockers (ARBs) or angiotensin-converting enzyme (ACE) inhibitors, and diuretics.

In addition, a propensity score-matched analysis was performed to reduce the bias in baseline characteristics between the low- and high-Zn groups. The following factors were included in a logistic regression model as covariates to calculate the propensity score: age; sex; BMI;

histories of DM and cardiovascular disease; ln(eGFR); ln(CRP); serum albumin levels; hemoglobin levels; dipstick proteinuria; and use of ARBs or ACE inhibitors. These factors were selected from baseline characteristics and the factors that were significantly different between the two groups, so that the population after matching would be as large as possible. A one-to-one match between the groups was obtained using nearest-neighbor matching replacement with a caliper width set at 0.25 of the standard deviation of the logit of the propensity score. Survival analysis was then conducted.

Interactions between Zn levels and baseline characteristics were evaluated using Cox proportional hazards models including baseline characteristics and the interaction terms of Zn levels × baseline characteristics [age, male sex, BMI, DM, cardiovascular disease, ln(eGFR), ln(CRP), serum albumin levels, hemoglobin levels, dipstick proteinuria, ARBs or ACE inhibitors, and diuretics]. Subgroup analysis was conducted for the statistically significant interaction terms.

Finally, to investigate the relationship between zinc supplementation and the prognosis of CKD, a survival analysis was conducted in two groups: patients who took a zinc-containing drug during the entire observation period and those who did not. The survival analysis was performed in both patients with low- and high-Zn levels. All statistical analyses were performed using SAS version 9.4 (SAS Institute, Cary, NC, USA) and R version 3.4.1 (R Foundation for Statistical Computing, Vienna, Austria). Statistical significance was defined as a two-sided $p < 0.05$.

## Results

### Baseline characteristics and outcomes

Of the 312 study patients, 160 were in the low-Zn group and 152 were in the high-Zn group (Fig 1). Their baseline characteristics are shown in Table 1. There were more patients with nephrotic syndrome in the low-Zn group than in the high-Zn group but there were no differences in the number of patients with DM, chronic liver disease, or bowel disease between the two groups. Patients with bowel disease were rare. In terms of medication, only eight patients overall were taking a zinc-containing drug at baseline. During the entire observation period, 94 patients took a zinc-containing drug, and the number was higher in the low-Zn group. Fewer patients took ARBs or ACE inhibitors and more took diuretics in the low-Zn group than in the high-Zn group. In laboratory measurements, the low-Zn group showed lower eGFR, serum albumin levels, and hemoglobin levels, and higher CRP levels than the high-Zn group.

Table 2 shows the incidence of outcomes. The primary outcomes were reached in 100 patients (32.1%) and exhibited higher incidence in the low-Zn group than in the high-Zn group (43.1% vs 20.4%, $p < 0.001$). The incidence of ESKD and that of death were higher in the low-Zn group. None of the patients with ESKD underwent kidney transplantation.

### Relationship between low serum zinc levels and outcomes

In Kaplan-Meier curves for the primary outcome, the survival rate was lower in the low-Zn group (log-rank test, $p < 0.001$; Fig 2). In addition, various Cox proportional hazards models adjusted for baseline characteristics showed a statistically significantly higher risk of the primary outcome in the low-Zn group than in the high-Zn group (Fig 3).

Furthermore, competing risks were assessed using Fine and Gray competing risk regression models. The risk of ESKD was higher in the low-Zn group than in the high-Zn group [adjusted HR 1.89 (95% CI 1.15, 3.10; $p = 0.012$)] but there was no difference in the risk of death ($p = 0.65$).

**Table 1. Baseline characteristics of patients grouped by Zn levels.**

| | All | Low-Zn group | High-Zn group | *p* |
|---|---|---|---|---|
| n | 312 | 160 | 152 | |
| **Demographic characteristic** | | | | |
| Age (years) | 69.5±14.4 | 69.8±14.4 | 69.2±14.4 | 0.72 |
| Male (%) | 165 (52.9) | 88 (55.0) | 77 (50.7) | 0.50 |
| BMI (kg/m$^2$) | 22.9±4.6 | 22.6±4.4 | 23.3±4.8 | 0.21 |
| **Comorbidity** | | | | |
| Hypertension (%) | 262 (84.0) | 132 (82.5) | 130 (85.5) | 0.54 |
| DM (%) | 138 (44.2) | 69 (43.1) | 69 (45.4) | 0.73 |
| Dyslipidemia (%) | 136 (43.6) | 59 (36.9) | 77 (50.7) | 0.017 |
| Cardiovascular disease (%) | 124 (39.7) | 68 (42.5) | 56 (36.8) | 0.36 |
| Nephrotic syndrome (%) | 47 (15.1) | 37 (23.1) | 10 (6.6) | <0.001 |
| Chronic liver disease (%) | 40 (12.8) | 22 (13.8) | 18 (11.8) | 0.74 |
| Bowel disease (%) | 4 (1.3) | 1 (0.6) | 3 (2.0) | 0.36 |
| Cancer (%) | 79 (25.3) | 41 (25.6) | 38 (25.0) | 1.0 |
| **Medication** | | | | |
| ARBs or ACE inhibitors (%) | 143 (45.8) | 64 (40.0) | 79 (52.0) | 0.041 |
| Diuretics (%) | 210 (67.3) | 121 (75.6) | 89 (58.6) | 0.002 |
| Zinc-containing drugs (baseline) (%) | 8 (2.6) | 1 (0.6) | 7 (4.6) | 0.033 |
| Zinc-containing drugs (whole period) (%) | 94 (30.1) | 62 (38.8) | 32 (21.1) | 0.001 |
| **Laboratory measurement** | | | | |
| Zn (μg/dl) | 59.6±14.6 | 48.2±8.4 | 71.6±9.3 | <0.001 |
| Creatinine (mg/dl) | 2.32 (1.36, 4.22) | 2.55 (1.50, 4.45) | 1.98 (1.25, 3.90) | 0.032 |
| eGFR (ml/min/1.73 m$^2$) | 20.3 (10.5, 35.1) | 18.2 (9.5, 33.3) | 22.7 (11.7, 38.3) | 0.046 |
| CRP (mg/dl) | 0.24 (0.09, 1.16) | 0.39 (0.12, 1.82) | 0.18 (0.08, 0.48) | <0.001 |
| Albumin (g/dl) | 3.2±0.8 | 2.8±0.8 | 3.6±0.6 | <0.001 |
| Hemoglobin (g/dl) | 9.8±1.9 | 9.3±1.7 | 10.3±1.9 | <0.001 |
| Dipstick proteinuria (≥1+) (%) | 216 (69.2) | 115 (71.9) | 101 (66.4) | 0.33 |

Continuous variables are shown as the mean ± SD or median (interquartile range). Categorical variables are shown as n (%). Abbreviations: Zn, serum zinc; BMI, body mass index; DM, diabetes mellitus; ARBs, angiotensin II receptor blockers; ACE, angiotensin-converting enzyme; eGFR, estimated glomerular filtration rate; CRP, C-reactive protein.

**Table 2. Incidence of outcomes in patients grouped by Zn levels.**

| | All | Low-Zn group | High-Zn group | *p* |
|---|---|---|---|---|
| n | 312 | 160 | 152 | |
| **Outcomes, n (%)** | | | | |
| Primary outcome | 100 (32.1) | 69 (43.1) | 31 (20.4) | <0.001 |
| ESKD | 74 (23.7) | 49 (30.6) | 25 (16.4) | 0.003 |
| Hemodialysis | 65 (20.8) | 45 (28.1) | 20 (13.2) | 0.001 |
| Peritoneal dialysis | 9 (2.9) | 4 (2.5) | 5 (3.3) | 0.75 |
| Death | 26 (8.3) | 20 (12.5) | 6 (3.9) | 0.007 |
| **Observation period (days)** | 365 (71, 365) | 163 (38, 365) | 365 (197, 365) | <0.001 |

Continuous variables are shown as the median (interquartile range). Categorical variables are shown as n (%). Abbreviations: Zn, serum zinc; ESKD, end-stage kidney disease.

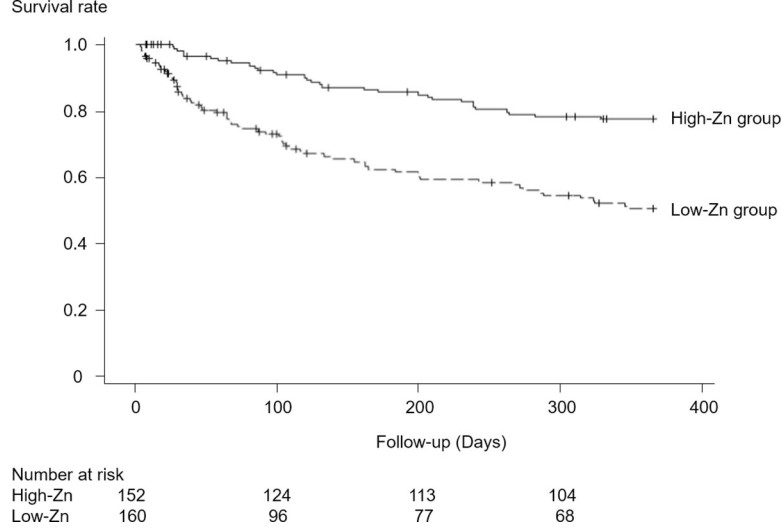

**Fig 2. Survival rates in the original study population.** Kaplan-Meier curves for the primary outcome over 1 year.

## Propensity score-matched analysis

Patients were matched in the two groups, with 87 patients included in each group. There were no statistically significant differences in patient characteristics between the two groups, except for Zn levels and the percentage of patients who took a zinc-containing drug during the entire observation period (S1 Table). The incidence of the primary outcome was higher in the low-Zn group than in the high-Zn group (S2 Table). In Kaplan-Meier curves for the primary outcome, the survival rate was lower in the low-Zn group (log-rank test, $p$ = 0.031; Fig 4). Moreover, in Cox proportional hazards models, the risk of the primary outcome was 1.81 times higher in the low-Zn group than in the high-Zn group (Table 3).

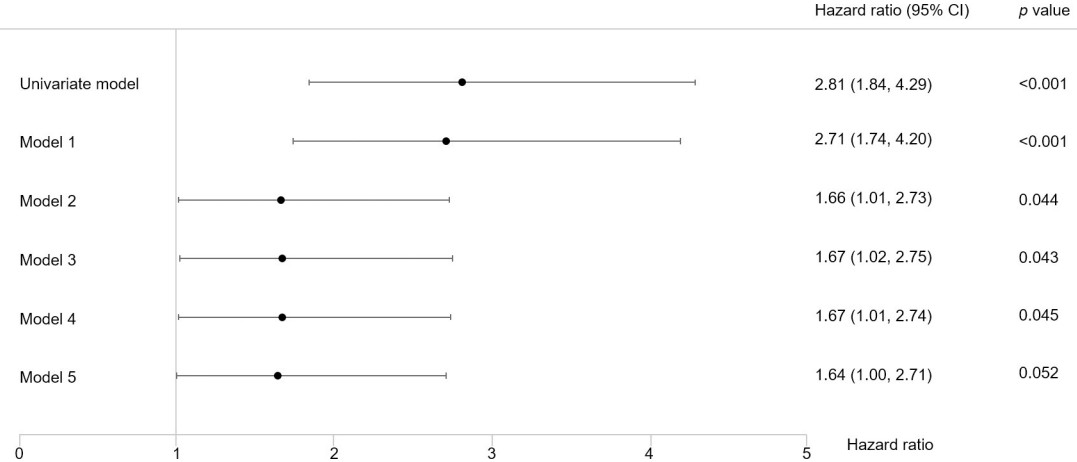

**Fig 3. Association of low Zn levels with the primary outcome.** The HRs of the low-Zn group to the high-Zn group are shown. Model 1 was adjusted for age, male sex, BMI, DM, cardiovascular disease, and ln(eGFR); Model 2, Model 1 + ln(CRP), serum albumin levels, and hemoglobin levels; Model 3, Model 2 + dipstick proteinuria; Model 4, Model 3 + ARBs or ACE inhibitors; Model 5, Model 4 + diuretics.

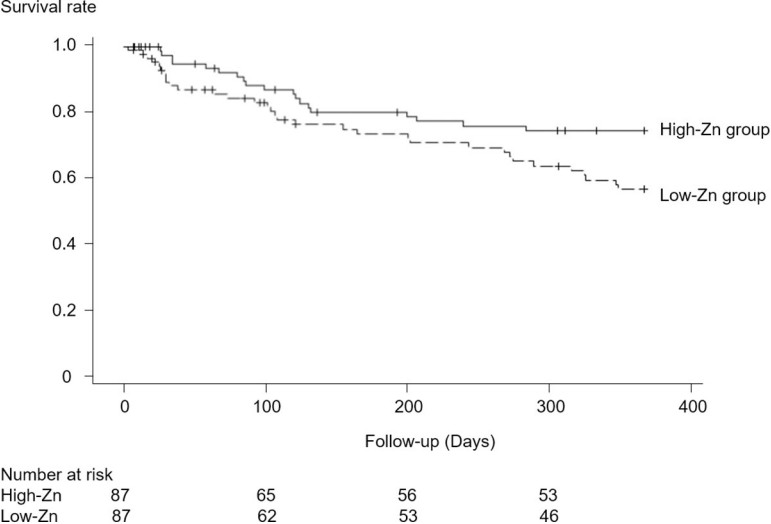

**Fig 4. Survival rates in the propensity score-matched population.** Kaplan-Meier curves for the primary outcome over 1 year.

In addition, Fine and Gray competing risk regression models showed higher risk of ESKD in the low-Zn group than in the high-Zn group [adjusted HR 2.76 (95% CI 1.43, 5.32; $p = 0.003$)] but no difference in the risk of death ($p = 0.44$).

## Effects of hypoalbuminemia on the relationship between Zn levels and outcome

The results of the evaluation of interaction terms with Zn level are shown in S3 Table. An interaction was observed between Zn and serum albumin levels ($p = 0.026$). Patients were next categorized into two subgroups using the baseline mean serum albumin level: those with low serum albumin levels (<3.2 g/dl; n = 137) and those with high serum albumin levels ($\geq$3.2 g/dl; n = 175). Kaplan-Meier curves for each subgroup are shown in Fig 5. In both subgroups, the survival rate for the primary outcome was lower in the low-Zn group (low-albumin, log-rank test, $p = 0.003$; high-albumin, log-rank test, $p = 0.020$). The results of analysis using a Cox proportional hazards model in each subgroup are shown in Table 4. In patients with low serum albumin levels, the risk of the primary outcome was 3.31 times higher in the low-Zn group than in the high-Zn group. In patients with high serum albumin levels, there was no difference in the primary outcome between the low- and high-Zn groups ($p = 0.52$).

## Relationship between zinc supplementation and the prognosis of CKD

S1 Fig shows the Kaplan-Meier curves of subgroups by Zn levels. In patients with low Zn levels (Zn level <60 μg/dl), patients taking a zinc-containing drug had a higher survival rate for the

**Table 3. Association of low Zn levels with the primary outcome in the propensity score-matched population.**

|  | **HR (95% CI)** | ***p*** |
|---|---|---|
| Univariate model | 1.84 (1.05, 3.24) | 0.034 |
| Adjusted model | 1.81 (1.02, 3.24) | 0.045 |

The HRs of the low-Zn group to the high-Zn group are shown. The adjusted model was adjusted for zinc-containing drugs (whole period).

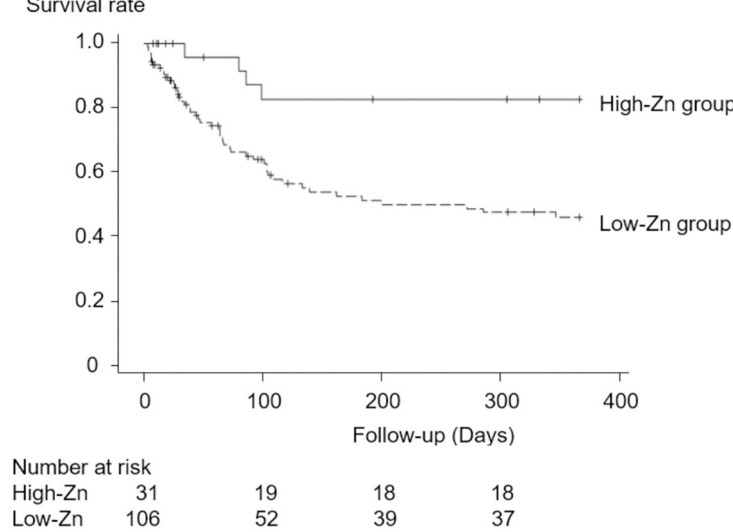

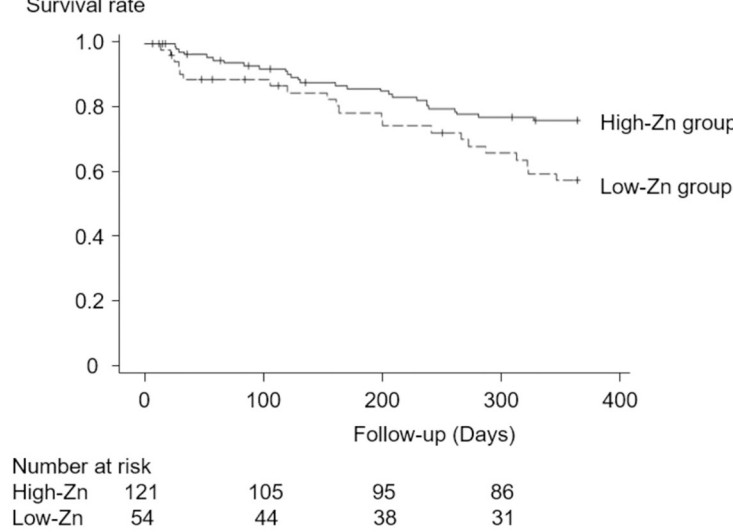

**Fig 5. Survival rates in the subgroups classified by serum albumin levels.** Kaplan-Meier curves for the primary outcome over 1 year. (A) Patients with low serum albumin levels ($<$3.2 g/dl). (B) Patients with high serum albumin levels ($\geq$3.2 g/dl). Survival rates were compared between the two groups by the log-rank test. The survival rate was lower in the low-Zn group than in the high-Zn group (A, log-rank test, $p$ = 0.003; B, log-rank test, $p$ = 0.020).

**Table 4. Association of Zn levels with the primary outcome in subgroups according to serum albumin levels.**

|  | Adjusted HR (95% CI) | $p$ |
|---|---|---|
| Low-albumin patients | 3.31 (1.13, 9.70) | 0.029 |
| High-albumin patients | 1.25 (0.63, 2.45) | 0.52 |

Adjusted HRs of the low-Zn group to the high-Zn group are shown. HRs were adjusted for baseline characteristics, such as age, male sex, BMI, DM, cardiovascular disease, ln(eGFR), ln(CRP), serum albumin levels, hemoglobin levels, dipstick proteinuria, ARBs or ACE inhibitors, and diuretics.

primary outcome (A, log-rank test, $p$ = 0.002). On the other hand, in patients with high Zn levels (Zn level ≥60 μg/dl), there was no difference in survival between the patients who were taking a zinc-containing drug and those were not (B, log-rank test, $p$ = 0.49). The results of analysis using the Cox proportional hazards model in patients with low- and high-Zn levels are shown in S4 Table. In patients with low Zn levels, the risk of the primary outcome was reduced by 62% in the patients taking a zinc-containing drug. However, in patients with high Zn levels, there was no difference in the risk of the primary outcome between the patients taking zinc-containing drugs and those not taking them ($p$ = 0.16).

## Discussion

In this study, in accordance with the guidelines of the Japanese Society of Clinical Nutrition [2], a serum zinc level of 60 μg/dl was used as the cut-off value to categorize patients into two groups. Survival analysis showed that zinc deficiency was a risk factor for the primary outcome, defined as ESKD or death. Analysis using competing risks models revealed that the low-Zn group had a high risk of ESKD. These results of different analyses confirmed the relationship between zinc deficiency and ESKD. Furthermore, in patients with low Zn levels, the risk of the primary outcome was lower in patients who took zinc-containing drugs during the observation period. To our knowledge, no clinical studies have investigated whether zinc deficiency is a risk factor for ESKD using hard renal endpoints.

Our results showed that zinc deficiency is a risk factor for ESKD. However, the mechanism underlying the relationship between zinc deficiency and renal dysfunction remains unclear. Various basic studies have shown that zinc is a regulator of oxidative stress [16–18]. Zinc is a cofactor of superoxide dismutase, which has antioxidant activity [19], and contributes to reducing oxidative stress. In addition, zinc deficiency has been shown to induce oxidative stress and renal damage via nicotinamide adenine dinucleotide phosphate (NADPH) oxidase [20]. A large body of evidence indicates that oxidative stress is the common denominator among the major pathways involved in the development and progression of kidney diseases [21]. NADPH oxidase has been identified as a major source of oxidative stress in kidney diseases [22, 23]. An intervention study in healthy individuals has also demonstrated that zinc supplementation reduces oxidative stress [24]. Moreover, zinc deficiency promotes renal fibroblast activation and leads to interstitial fibrosis in diabetic mice [25]. In addition to inhibiting the progression of kidney fibrosis, zinc supplementation improves liver fibrosis in patients with early cirrhosis [26]. These reports indicate an association between zinc deficiency and organ damage due to fibrosis. Thus, zinc deficiency may affect kidney function via oxidative stress and fibrosis.

On the other hand, the main causes of zinc deficiency are zinc excretion, insufficient intake, and malabsorption [27]. Increased zinc excretion causes zinc deficiency in patients with kidney diseases as well as in those on hemodialysis [8–10]. Compared with non-CKD patients, CKD patients have higher urinary zinc excretion, which tends to increase as the CKD stage progresses [8]. Insufficient zinc intake is more likely to occur in elderly people with malnutrition, particularly those with CKD [28]. Inadequate zinc intake causes serum zinc levels to fall quickly [29]. For example, in healthy men, Zn levels fall by 65% on average after 5 weeks on a zinc-restricted diet [30]. Moreover, gastrointestinal absorption is lower in CKD patients [31]. Accordingly, CKD patients are prone to zinc deficiency for various reasons.

Furthermore, in this study, an interaction was observed between Zn and serum albumin levels. The results of subgroup analysis suggested that hypoalbuminemia may exacerbate CKD progression induced by zinc deficiency. About 80% of serum zinc is bound to albumin [5, 32], and serum zinc and albumin levels are positively correlated [33]. Malnourished patients, who

often have hypoalbuminemia, are prone to zinc deficiency. Moreover, hypoalbuminemia reduces the amount of zinc bound to albumin in the blood, leading to increased urinary excretion of zinc. In this study, the number of nephrotic syndrome patients was statistically significantly higher in the low-Zn group. Thus, higher urinary albumin excretion further decreases Zn levels [9]. Taken together, these findings suggest that CKD patients with hypoalbuminemia are at high risk of zinc deficiency.

Considering that hypoalbuminemia is a well-defined risk factor for CKD progression [34, 35], it can be said that hypoalbuminemia affects both zinc deficiency and CKD progression. Therefore, because CKD patients with hypoalbuminemia tend to have low Zn levels, which lead to the progression of CKD, a vicious cycle can occur in the relationship between low Zn levels and CKD progression. Periodic monitoring of serum zinc and albumin levels is important as a strategy to prevent CKD progression.

There are some limitations to this study. First, because this is an observational study, the possibility of reverse causality cannot be completely ruled out. Therefore, we analyzed data using propensity score matching to minimize the bias of baseline characteristics. Second, we were unable to collect data over time, because Zn levels were not regularly measured in daily clinical practice. Third, the data set given to us in this study may not have enough data on confounders. Indeed, there was a lack of data on medications such as erythropoiesis-stimulating agents and immunosuppressant drugs, quantitative proteinuria, management of hypertension and diabetes, and lifestyle habits such as smoking and diet. Finally, because the participants of this study were likely to be in a condition requiring measurement of Zn levels, there might be a selection bias. Future intervention studies are required to confirm the causal relationship between zinc deficiency and renal prognosis.

In conclusion, this study showed that zinc deficiency is a risk factor for CKD progression. In addition, an interaction was observed between Zn and serum albumin levels. Therefore, Zn levels should be measured routinely and corrected in CKD patients, especially those with hypoalbuminemia.

## Supporting information

**S1 Table. Characteristics of patients after propensity score matching.**
(PDF)

**S2 Table. Incidence of outcomes in patients grouped by Zn levels after propensity score matching.**
(PDF)

**S3 Table. Interaction between Zn levels and baseline characteristics.**
(PDF)

**S4 Table. Association of zinc-containing drugs with the primary outcome in subgroups according to Zn levels.**
(PDF)

**S1 Fig. Survival rates in the subgroups classified by Zn levels.** Kaplan-Meier curves for the primary outcome over 1 year. (A) Patients with low Zn levels (Zn level <60 μg/dl). (B) Patients with high Zn levels (Zn level ≥60 μg/dl). Survival rates were compared between the two groups (with and without zinc-containing drugs) by the log-rank test.
(TIF)

## Author Contributions

**Conceptualization:** Atsuyuki Tokuyama, Eiichiro Kanda, Seiji Itano.

**Data curation:** Atsuyuki Tokuyama, Eiichiro Kanda, Seiji Itano.

**Formal analysis:** Atsuyuki Tokuyama, Eiichiro Kanda, Seiji Itano.

**Funding acquisition:** Eiichiro Kanda.

**Investigation:** Atsuyuki Tokuyama, Eiichiro Kanda, Seiji Itano.

**Methodology:** Atsuyuki Tokuyama, Eiichiro Kanda, Seiji Itano.

**Project administration:** Atsuyuki Tokuyama, Eiichiro Kanda, Seiji Itano.

**Resources:** Atsuyuki Tokuyama, Eiichiro Kanda, Seiji Itano.

**Software:** Atsuyuki Tokuyama, Eiichiro Kanda, Seiji Itano.

**Supervision:** Naoki Kashihara.

**Validation:** Megumi Kondo, Yoshihisa Wada, Hiroyuki Kadoya, Kengo Kidokoro, Hajime Nagasu, Tamaki Sasaki, Naoki Kashihara.

**Writing – original draft:** Atsuyuki Tokuyama, Eiichiro Kanda, Seiji Itano.

**Writing – review & editing:** Atsuyuki Tokuyama, Eiichiro Kanda, Seiji Itano.

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
