## [Decision Letter · Decision Letter 0]

5 Mar 2021

PONE-D-21-03589

Effect of zinc deficiency on chronic kidney disease progression: A retrospective cohort study

PLOS ONE

Dear Dr. Kanda,

Thank you for submitting your manuscript to PLOS ONE. After careful consideration, we feel that it has merit but does not fully meet PLOS ONE’s publication criteria as it currently stands. Therefore, we invite you to submit a revised version of the manuscript that addresses the points raised during the review process.

Please respond to the additional editor comments as well as the reviewer's comments.

We look forward to receiving your revised manuscript.

Kind regards,

Kojiro Nagai

Academic Editor

PLOS ONE

Journal Requirements:

Additional Editor Comments:

The authors investigated the relationship between Zn deficiency and the progression of CKD.

I understand the novelty of this study. However, I wonder if the risk of Zn deficiency was overestimated, because there seems to be no quantitative evaluation of proteinuria in this study, which is one of the well-known risk factors for CKD progression. Therefore, I suggest to add the analysis of one more Cox proportional hazards model adjusted for age, male, BMI, DM, cardiovascular disease, ln(eGFR), ln(CRP), serum albumin levels, hemoglobin levels (deletion of dipstick proteinuria from model 3), because I guess that the simple estimation of proteinuria in this study may affect the results.

In addition, there are many factors to influence the progression of CKD such as the intake of immunosuppressant drugs and the management of hypertension and diabetes during the observational period. It is better to add those data. If they are not available, it should be noted in the limitation of this study.

Reviewers' comments:

Reviewer's Responses to Questions

**Comments to the Author**

1. Is the manuscript technically sound, and do the data support the conclusions?

Reviewer #1: Yes

Reviewer #2: Yes

Reviewer #3: Yes

2. Has the statistical analysis been performed appropriately and rigorously? 

Reviewer #1: Yes

Reviewer #2: Yes

Reviewer #3: Yes

3. Have the authors made all data underlying the findings in their manuscript fully available?

Reviewer #1: Yes

Reviewer #2: Yes

Reviewer #3: Yes

4. Is the manuscript presented in an intelligible fashion and written in standard English?

Reviewer #1: Yes

Reviewer #2: Yes

Reviewer #3: Yes

5. Review Comments to the Author

Reviewer #1: General comments

This manuscript reported the effect of zinc deficiency on the decline in kidney function in CKD patients from single-center, retrospective cohort study. This study contains some novel information, however, there are some concerns that should be discussed.

Comments

1. Was dietary guidance, such as cocoa, given to the low zinc group?

2. Were there any cases in which the changes in zinc level could be followed? If there were any cases, did changes in zinc level affect the prognosis?

3. The authors need to evaluate the bias of measuring zinc levels. Is the timing of the zinc measurement outpatient? Or during hospitalization for some event?

4. Is the association between zinc levels and hypoalbuminemia due to proteinuria or due to low nutrition?

Reviewer #2: The authors retrospectively examined the association between Zn levels and the hard outcome of ESKD or death using historical electronic medical records. The results suggest that low Zn is a significant risk and a poor prognosis, especially in the hypoalbuminemic group. This study is thought to be suggestive in terms of clinical management and nutrition of CKD patients. However, there are several points of concern that we would like to mention below.

Comments to the author

1) The low Zn group has significantly worse nephrotic syndrome, hypoalbuminemia, dyslipidemia, impaired renal function, and anemia. In fact, in Fig. 3, the significance of the results was reduced when ln(CRP), serum albumin, hemoglobin levels and dipstick proteinuria were added to the correction, suggesting that these factors may be associated with prognosis. Based on this, the authors conducted a propensity score-matched analysis. The authors need to provide the reason why they selected the baseline characteristics [age, sex, BMI, DM, cardiovascular disease, ln(eGFR), ln(CRP), serum albumin levels, hemoglobin levels, dipstick proteinuria, and use of angiotensin II receptor blockers (ARBs) or angiotensin-converting enzyme (ACE) inhibitors] for adjustment in regression models and calculation of propensity score.　If there were unmeasured confounders such as use of ESA, please state in the limitation section.

2) As with RAS inhibitors, diuretics, which have a significant impact on renal function trends, should be added as a correction factor.

3) Since hypoalbuminemia due to nephrotic syndrome can lead to both decreased Zn and decreased renal function, the possibility that hypoalbuminemia is a bystander cannot be denied. For greater reliability, it is better that urine protein quantification would be included in the analysis. If urine protein quantification is not sufficiently available, this should be noted in the limitation.

4) As the authors mentioned in the DISCUSSION, the most important point is whether Zn supplementation is effective or not. How many of the patients in this study were newly started on zinc during the observation period? What is the relationship between new initiation of zinc and prognosis?

Reviewer #3: Tokuyama et al. reported a retrospective cohort study that clarified zinc deficiency as a risk of CKD progression in patients with CKD, in which authors used cox proportional hazards models and propensity score-matched analysis. In addition, authors showed the association between serum zinc levels and albumin levels, and concluded hypoalbuminemia affected the relationship between zinc and outcomes. This study and statistical analysis were well designed, and clearly showed the effect of zinc deficiency on CKD progression.

Major comment

(1) In figure 6, authors presented the effect of zinc deficiency on outcome in low-albumin patients and high-albumin patients in which concluded no association between zinc deficiency and outcome in high albumin patients. However, it is not clear why there is the difference of zinc-associated outcome between both groups. Authors should describe the possible mechanisms or hypothesis that zinc deficiency affect the outcome in low-albumin groups.

(2) A title is “Effect of zinc deficiency on chronic kidney disease progression”. If there is no association of zinc deficiency and outcome in CKD patients with high albumin group, a title may mention the effect of hypoalbuminemia on the relationship between zinc deficiency and outcome.

6. PLOS authors have the option to publish the peer review history of their article (what does this mean?). If published, this will include your full peer review and any attached files.

Reviewer #1: No

Reviewer #2: No

Reviewer #3: No

---

## [Author Response · Author response to Decision Letter 0]

19 Apr 2021

Dear Editor and Reviewers,

Thank you very much for reviewing our manuscript. We appreciate the valuable advice offered. We have addressed the comments and revised the manuscript accordingly. Please find our point-by-point responses below.

Reply to Additional Editor Comments

Additional Editor Comments:

The authors investigated the relationship between Zn deficiency and the progression of CKD.

I understand the novelty of this study. However, I wonder if the risk of Zn deficiency was overestimated, because there seems to be no quantitative evaluation of proteinuria in this study, which is one of the well-known risk factors for CKD progression. Therefore, I suggest to add the analysis of one more Cox proportional hazards model adjusted for age, male, BMI, DM, cardiovascular disease, ln(eGFR), ln(CRP), serum albumin levels, hemoglobin levels (deletion of dipstick proteinuria from model 3), because I guess that the simple estimation of proteinuria in this study may affect the results.

In addition, there are many factors to influence the progression of CKD such as the intake of immunosuppressant drugs and the management of hypertension and diabetes during the observational period. It is better to add those data. If they are not available, it should be noted in the limitation of this study.

Response: Quantitative proteinuria could not be assessed in this study due to abundant missing data. We have added this aspect as a study limitation to the Discussion section (p 16, line 277-280). As you suggested, we created two Cox hazards models (Fig 3), one without dipstick proteinuria [Model 2: age, male sex, BMI, DM, cardiovascular disease, ln(eGFR), ln(CRP), serum albumin levels, hemoglobin levels], and the other with dipstick proteinuria added to Model 2 [Model 3: age, male sex, BMI, DM, cardiovascular disease, ln(eGFR), ln(CRP), serum (CRP), serum albumin levels, hemoglobin levels, and dipstick proteinuria]. The hazard ratio was almost the same in the two models.

In addition, data on immunosuppressant drugs and the management of hypertension and diabetes were not adequately included in the data set given to us and could not be assessed. Therefore, the confounding factors affecting the progression of CKD may not have been sufficiently covered, and we have thus included this aspect as a limitation of our study.

Reply to Reviewer #1

1. Was dietary guidance, such as cocoa, given to the low zinc group?

Response: The data set given to us lacked lifestyle data, including dietary content, and it was unclear whether the low-Zn group was given dietary guidance, such as that regarding cocoa consumption. Therefore, we have added this aspect as a study limitation to the Discussion section (p 16, line 277-280).

2. Were there any cases in which the changes in zinc level could be followed? If there were any cases, did changes in zinc level affect the prognosis?

Response: The number of cases whose serum zinc levels could be followed over time was not large enough to properly examine the relationship between the changes in serum zinc levels and prognosis. However, we were able to determine the number of patients who took a zinc-containing drug during the entire observation period and have added this information to Table 1. We have thus added an analysis of the prognosis of patients who used a zinc-containing drug during the entire observation period and those who did not (p 6, line 108-111). The results showed that, in patients with low serum zinc levels, the risk of the primary outcome was lower in patients who took a zinc-containing drug (p 13, line 210-221). These results have been added as S1 Figure and S4 Table.

3. The authors need to evaluate the bias of measuring zinc levels. Is the timing of the zinc measurement outpatient? Or during hospitalization for some event?

Response: The serum zinc level was measured in either an outpatient or inpatient setting, but we were unable to distinguish between the two. Indeed, the zinc measurement was likely to be performed during prolonged anemia or when patients showed clinical symptoms of suspected zinc deficiency. Because there may be a selection bias among CKD patients in this regard, it has been mentioned in the limitations (p 16, line 280-281).

4. Is the association between zinc levels and hypoalbuminemia due to proteinuria or due to low nutrition?

Response: In terms of baseline patient characteristics, nephrotic syndrome was significantly more common in the low-Zn group. Dipstick proteinuria was also more common in the low-Zn group, although not significantly so. Therefore, proteinuria may be related in no small part to the relationship between serum zinc levels and hypoalbuminemia.

Reply to Reviewer #2

1. The low Zn group has significantly worse nephrotic syndrome, hypoalbuminemia, dyslipidemia, impaired renal function, and anemia. In fact, in Fig. 3, the significance of the results was reduced when ln(CRP), serum albumin, hemoglobin levels and dipstick proteinuria were added to the correction, suggesting that these factors may be associated with prognosis. Based on this, the authors conducted a propensity score-matched analysis. The authors need to provide the reason why they selected the baseline characteristics [age, sex, BMI, DM, cardiovascular disease, ln(eGFR), ln(CRP), serum albumin levels, hemoglobin levels, dipstick proteinuria, and use of angiotensin II receptor blockers (ARBs) or angiotensin-converting enzyme (ACE) inhibitors] for adjustment in regression models and calculation of propensity score.　If there were unmeasured confounders such as use of ESA, please state in the limitation section.

Response: As adjustment factors for the regression analysis, in addition to basic patient characteristics, we preferentially selected factors that were significantly different between the two groups or that could affect the outcomes. We also selected factors for calculating the propensity scores from basic patient characteristics, as well as factors that were significantly different between the two groups (low- and high-Zn groups) at baseline, to obtain as large a population after matching as possible. These statements have been added to the manuscript (p 6, line 88-93 and line 98-100).

Unmeasured confounders, such as the use of ESA, have been added as a study limitation to the Discussion section (p 16, line 277-280).

2. As with RAS inhibitors, diuretics, which have a significant impact on renal function trends, should be added as a correction factor.

Response: As you point out, diuretics are an important factor affecting renal function, and we have added their use to the regression analyses, both the Cox hazards models and the Fine and Gray models.

3. Since hypoalbuminemia due to nephrotic syndrome can lead to both decreased Zn and decreased renal function, the possibility that hypoalbuminemia is a bystander cannot be denied. For greater reliability, it is better that urine protein quantification would be included in the analysis. If urine protein quantification is not sufficiently available, this should be noted in the limitation.

Response: Quantitative proteinuria could not be assessed in this study due to numerous missing data. This aspect has been added as a study limitation to the Discussion section (p 16, line 277-280).

4. As the authors mentioned in the DISCUSSION, the most important point is whether Zn supplementation is effective or not. How many of the patients in this study were newly started on zinc during the observation period? What is the relationship between new initiation of zinc and prognosis?

Response: In total, 94 patients took a zinc-containing drug during the entire observation period. This information has been added to Table 1. We have also added an analysis of the prognosis of patients who used a zinc-containing drug and those who did not. The results showed that, in patients with low serum zinc levels, the risk of the primary outcome was lower in patients who took a zinc-containing drug (p 13, line 210-221). These results have been added as S1 Figure and S4 Table.

Reply to Reviewer #3

1. In figure 6, authors presented the effect of zinc deficiency on outcome in low-albumin patients and high-albumin patients in which concluded no association between zinc deficiency and outcome in high albumin patients. However, it is not clear why there is the difference of zinc-associated outcome between both groups. Authors should describe the possible mechanisms or hypothesis that zinc deficiency affect the outcome in low-albumin groups.

Response: We believe that your comment raises a very important point. As we mentioned in the Discussion section, hypoalbuminemia can affect both zinc deficiency and CKD progression. Therefore, we believe that, because CKD patients with hypoalbuminemia tend to have low Zn levels, which lead to the progression of CKD, a vicious cycle can occur in the relationship between low Zn levels and CKD progression. However, the detailed mechanism has not been clarified yet and is a topic for further investigation.

2. A title is “Effect of zinc deficiency on chronic kidney disease progression”. If there is no association of zinc deficiency and outcome in CKD patients with high albumin group, a title may mention the effect of hypoalbuminemia on the relationship between zinc deficiency and outcome.

Response: In response to your comment, the title has been changed to "Effect of zinc deficiency on chronic kidney disease progression and effect modification by hypoalbuminemia" to take into account the effect of hypoalbuminemia on the relationship between zinc deficiency and outcome.

Once again, we thank you for your kind comments regarding our manuscript.

---

## [Decision Letter · Decision Letter 1]

28 Apr 2021

Effect of zinc deficiency on chronic kidney disease progression and effect modification by hypoalbuminemia

PONE-D-21-03589R1

Dear Dr. Kanda,

We’re pleased to inform you that your manuscript has been judged scientifically suitable for publication and will be formally accepted for publication once it meets all outstanding technical requirements.

Kind regards,

Kojiro Nagai

Academic Editor

PLOS ONE

Additional Editor Comments (optional):

Reviewers' comments:

Reviewer's Responses to Questions

**Comments to the Author**

1. If the authors have adequately addressed your comments raised in a previous round of review and you feel that this manuscript is now acceptable for publication, you may indicate that here to bypass the “Comments to the Author” section, enter your conflict of interest statement in the “Confidential to Editor” section, and submit your "Accept" recommendation.

Reviewer #1: All comments have been addressed

Reviewer #2: All comments have been addressed

Reviewer #3: All comments have been addressed

2. Is the manuscript technically sound, and do the data support the conclusions?

Reviewer #1: Yes

Reviewer #2: Partly

Reviewer #3: Yes

3. Has the statistical analysis been performed appropriately and rigorously? 

Reviewer #1: Yes

Reviewer #2: Yes

Reviewer #3: Yes

4. Have the authors made all data underlying the findings in their manuscript fully available?

Reviewer #1: Yes

Reviewer #2: Yes

Reviewer #3: Yes

5. Is the manuscript presented in an intelligible fashion and written in standard English?

Reviewer #1: Yes

Reviewer #2: (No Response)

Reviewer #3: Yes

6. Review Comments to the Author

Reviewer #1: In this manuscript, the authors revised their manuscript in accordance with our review. This manuscript fulfilled our suggestion.

Reviewer #2: (No Response)

Reviewer #3: The manuscript has been revised accordingly, and I have no further comments to be addressed for the revized manuscript.

7. PLOS authors have the option to publish the peer review history of their article (what does this mean?). If published, this will include your full peer review and any attached files.

Reviewer #1: No

Reviewer #2: No

Reviewer #3: No

---

## [Editor Report · Acceptance letter]

3 May 2021

PONE-D-21-03589R1 

Effect of zinc deficiency on chronic kidney disease progression and effect modification by hypoalbuminemia 

Dear Dr. Kanda:

I'm pleased to inform you that your manuscript has been deemed suitable for publication in PLOS ONE. Congratulations! Your manuscript is now with our production department. 

Kind regards, 

on behalf of

Dr. Kojiro Nagai 

Academic Editor

PLOS ONE